# Image-Guided Brachytherapy for Salvage Reirradiation: A Systematic Review

**DOI:** 10.3390/cancers13061226

**Published:** 2021-03-11

**Authors:** Sophie Bockel, Sophie Espenel, Roger Sun, Isabelle Dumas, Sébastien Gouy, Philippe Morice, Cyrus Chargari

**Affiliations:** 1Department of Radiation Oncology, Gustave Roussy, Paris-Saclay University, 94800 Villejuif, France; sophie.espenel@gustaveroussy.fr (S.E.); roger.sun@gustaveroussy.fr (R.S.); Isabelle.dumas@gustaveroussy.fr (I.D.); cyrus.chargari@gustaveroussy.fr (C.C.); 2Department of Brachytherapy, Gustave Roussy, Paris-Saclay University, 94800 Villejuif, France; 3Institut National de la Santé et de la Recherche Médicale (INSERM) U1030, Molecular Radiotherapy, Paris-Saclay University, 94800 Villejuif, France; 4Department of Surgery, Gustave Roussy, Paris-Saclay University, 94800 Villejuif, France; Sebastien.gouy@gustaveroussy.fr (S.G.); philippe.morice@gustaveroussy.fr (P.M.); 5French Military Health Academy, Ecole du Val-de-Grâce, 75015 Paris, France

**Keywords:** radiotherapy, brachytherapy, cervical cancer, vaginal cancer, endometrial cancer, vulvar cancer, gynecologic cancer, reirradiation

## Abstract

**Simple Summary:**

Local recurrence in gynecological malignancies occurring in a previously irradiated field is a difficult clinical issue. Curative-intent treatment is salvage surgery and is associated with non-negligible peri-operative morbidity and has a substantial impact on long-term quality of life. Reirradiation, using three-dimensional image-guided brachytherapy (3D-IGBT), might be a suitable alternative, especially in non-operable patients. The aim of this review is to report outcomes and toxicities of reirradiation 3D-IGBT in this context. 3D-IGBT appears to be a feasible alternative to salvage surgery in inoperable patients, with an acceptable outcome for patients who have no other curative therapeutic options, however long-term toxicities were high in some studies. Each case should be referred to highly experienced expert centers.

**Abstract:**

Background: Local recurrence in gynecological malignancies occurring in a previously irradiated field is a challenging clinical issue. The most frequent curative-intent treatment is salvage surgery. Reirradiation, using three-dimensional image-guided brachytherapy (3D-IGBT), might be a suitable alternative. We reviewed recent literature concerning 3D-IGBT for reirradiation in the context of local recurrences from gynecological malignancies. Methods: We conducted a large-scale literature research, and 15 original studies, responding to our research criteria, were finally selected. Results: Local control rates ranged from 44% to 71.4% at 2–5 years, and overall survival rates ranged from 39.5% to 78% at 2–5 years. Grade ≥3 toxicities ranged from 1.7% to 50%, with only one study reporting a grade 5 event. Results in terms of outcome and toxicities were highly variable depending on studies. Several studies suggested that local control could be improved with 2 Gy equivalent doses >40 Gy. Conclusion: IGBT appears to be a feasible alternative to salvage surgery in inoperable patients or patients refusing surgery, with an acceptable outcome for patients who have no other curative therapeutic options, however at a high cost of long-term grade ≥3 toxicities in some studies. We recommend that patients with local recurrence from gynecologic neoplasm occurring in previously irradiated fields should be referred to highly experienced expert centers. Centralization of data and large-scale multicentric international prospective trials are warranted. Efforts should be made to improve local control while limiting the risk of toxicities.

## 1. Introduction

Local recurrence in advanced gynecologic malignancies is not so unusual, and remains a highly challenging clinical issue, especially when developed in a previously irradiated field. Indeed, the local relapse rate of cervical cancer ranges from 10% to 30%, when treated with upfront surgery, or definitive chemotherapy plus radiotherapy followed by brachytherapy [1,2]. Recent data showed a significant decrease in local relapse probability with dose escalation, however some patients may still experience a relapse within a previously irradiated area, or in field borders [3]. In most cervical cancers, local relapses occurring after definitive chemoradiation are located in the distal vagina and/or in the parametria/pelvic sidewall, in areas receiving less than 80–85 Gy [4]. Regarding primary endometrial cancer, patients with limited stage disease have a favorable prognosis, however local recurrences are still observed, up to 5%, despite adjuvant treatment [5]. Most local recurrences occur at the vaginal vault, which is commonly within an area of prior radiation (postoperative brachytherapy ± external radiotherapy) [5]. Concerning vaginal cancer, a study from the MD Anderson Cancer Centre, including 193 patients, reported pelvic disease control rates of 86% for Stage I, 84% for Stage II, and 71% for Stage III–IVA at 5 years, with a predominant pattern of locoregional relapse after definitive radiation therapy [6]. Lastly, for vulvar carcinoma, isolated local recurrence occurs in 20% to 23%, depending on the studies, of vulvar cancer after local treatment, including radiation therapy [7]. In addition to the situations of local recurrence, some patients with Human Papillomavirus-related disease may develop second primary tumors. This is also a difficult situation when second primary tumors occur in a previously irradiated area (this situation is not discussed here, however, some challenges are somewhat similar).

The most frequently proposed curative-intent treatment in patients with recurrent or persistent pelvic cancer in a previously irradiated area is surgery, usually based on anterior, posterior, or total pelvectomy. Performing salvage surgery for a local relapse in sites previously irradiated is technically challenging, with a high perioperative morbidity rate [8,9,10,11,12] and substantial impact on long-term quality of life [13]. Thus, salvage surgery can only be offered to a very select group of patients with central pelvic recurrence and cannot be offered in frail patients (elderly and/or with comorbidities). Chemotherapy plays a palliative role in this subset of patients [14], with a median overall survival (OS) of 17.5 months [15]. Unfortunately, local recurrences in previously irradiated areas and in patients who previously received one or more chemotherapy regimens are rarely chemo-sensitive [16].

Reirradiation may be an alternative option, especially in non-operable patients and/or for patients refusing pelvic exenteration, in order to preserve the structures and functions of the pelvic organs, avoiding surgical sequelae. To date, only scarce data based on retrospective series are available in this setting and no clear recommendation can guide irradiation technique or dose.

External beam radiation therapy (EBRT) is usually not delivered at effective doses because of the high risk of complications [17]. Based on its ability to deliver a highly conformal dose, stereotactic body radiation therapy (SBRT) was evaluated in pelvic recurrence in previously irradiated areas, however only in retrospective studies [17,18,19]. To date, dosimetric comparisons between brachytherapy and SBRT show that SBRT delivers higher doses to the organ-at-risk (OARs), thus carrying a higher risk of toxicity [20,21,22]. This is in line with data from the national cancer database of radiation therapy, showing that modern EBRT techniques cannot replace brachytherapy to achieve both focal dose escalation and organs at risk sparing [11,23].

In light of these considerations, interstitial brachytherapy seems particularly suitable for the management of central and paracentral pelvic recurrence because of the accessibility through the transperineal route and better conformality than EBRT and SBRT techniques. In the last decade, brachytherapy has benefited from major technological improvements, with three-dimensional (3D) image-guided brachytherapy (IGBT). The use of computed tomography (CT) and magnetic resonance imaging (MRI) for treatment planning allows a precise delineation of the tumor and of OARs, improving target dose coverage. Compared to two-dimensional (2D) brachytherapy, 3D-IGBT allows the delivery of higher doses to the tumor, while minimizing doses to normal tissue [24]. In cervical cancer, these improvements lead to a better outcome, both in terms of local rates and toxicity probability [25]. However, in the context of reirradiation, there is considerable variability in patient selection, dose and fractionation, and brachytherapy techniques. Furthermore, the potential toxicity of reirradiation is a major concern, requiring special attention in the treatment planning process.

We review recent literature on interstitial 3D-IGBT reirradiation as a salvage treatment in locally recurrent pelvic malignancies. Possibilities to perform salvage reirradiation in this highly challenging clinical setting are reviewed, in light of published literature.

## 2. Materials and Methods

### 2.1. Data Sources and Search Strategy

For this exhaustive literature review, we searched literature on 2 November 2020 to identify published articles that reported local recurrences from gynecologic cancers, occurring in previously irradiated fields, treated by 3D-IGBT, from 2000 to 2020. We did a systematic literature search on PubMed and Google Scholar, using the following search terms and boolean operators: (“pelvic recurrence” OR “vaginal recurrence” OR “vulvar recurrence” OR “recurrent pelvic malignancies”) AND (“gynecologic neoplasm” OR “gynecologic malignancies” OR “gynecologic cancer”) AND “brachytherapy” AND “reirradiation”. The search was updated on 13 February 2021.

Through this procedure, we found 45 studies on PubMed and 123 studies on Google Scholar. The duplicated studies were removed, and article titles were then evaluated. Abstracts found to be relevant to the topic of interest were shortlisted. Then, full-length papers of the shortlisted articles were assessed for the eligibility criteria. The articles that fulfilled the criteria were finally selected for the systematic review.

### 2.2. Study Selection

The inclusion criteria were as follows: (1) studies reporting patients with local recurrence from gynecologic cancer treated with 3D-IGBT (High Dose Rate (HDR), Pulse Dose Rate (PDR), or Low Dose Rate (LDR) BT) in a reirradiation setting, (2) all type of studies (including randomized controlled trials, prospective, retrospective, and case series) comprising more than 5 patients, (3) studies written in English, and (4) studies published from January 2000 to December 2020.

Exclusion criteria were: (1) Studies that did not specifically report brachytherapy reirradiation outcome/toxicity/dosimetric data, and (2) studies with exclusive two-dimensional treatment planning brachytherapy.

### 2.3. Data Extraction and Quality Assessment

A review protocol was defined according to the Preferred Reporting Items for Systematic Review and Meta-Analyses (PRIMSA) guidelines [26] (Appendix A), and studies that met the inclusion criteria using the PICOS (population, intervention, comparison, outcome, and setting) framework [27] were selected (Figure 1).

Shortlisted abstracts found to be relevant to the topic of interest and full-length articles fulfilling inclusion criteria were reviewed independently by two authors (S.B. and S.E.) and any disagreement was resolved by consensus with a third author (C.C.). Reasons for excluding studies were documented. The following data were extracted: First author, date of publication, number of patients included, number of patients treated with reirradiation brachytherapy, type of primary disease, type of recurrence, previous treatment modality, treatment-free interval, recurrent tumor volume, brachytherapy technique, dosimetric parameters, median follow-up, outcome, and toxicities.

## 3. Results

The literature search resulted in 168 articles (45 through PubMed and 123 through Google Scholar). After removing duplicate, not written in English, and off-topic studies, 44 full-length papers were assessed for eligibility criteria. Five review articles were found [17,19,28,29,30], and their references were crossed-searched for eventual additional studies. In some cases, we found articles combining patients with and without a history of prior radiation therapy. When data concerning re-irradiated patients were not separately available, the article was excluded from analysis (*n* = 14), as well as series with less than five patients with recurrent gynecologic malignancies re-irradiated with brachytherapy (*n* = 3). Papers with treatment planning performed using exclusively two-dimensional techniques were also excluded (*n* = 7). Finally, 15 total original research articles were included in our final analysis (Figure 1).

### 3.1. General Aspects

Over the past 20 years, 14 retrospectives series, and one phase II prospective study with only few patients [31], reported clinical experiences of high dose rate (HDR) or low dose rate (LDR) 3D-IGBT in previously irradiated pelvic recurrences of gynecologic malignancies (Table 1) [31,32,33,34,35,36,37,38,39,40,41,42,43,44,45]. In nine of those, 100% of the population received prior BT, prior EBRT, or a combination of EBRT followed with BT [31,32,33,36,40,41,43,44,45]. In the six remaining articles, only a variable subset of patients (from 10% to 85.7% of the studied population) received IGBT in a re-irradiation setting [34,35,37,38,39,42].

Regarding the size of the population, the median number of included patients receiving re-irradiation BT was 20 (range, 6–52), with 334 total re-irradiated patients considered in all studies included. The three largest series were published by Mabuchi et al. in 2014, Feddock et al. in 2017, and Mahantshetty et al. in 2014, with 52, 42, and 30 patients respectively, undergoing IGBT for reirradiation [33,40,45].

The median follow-up was available in 12/14 studies (not reported in two [36,37]) and ranged from 16.3 to 55.6 months (median, 27.6 months).

### 3.2. Patient’s Selection and Population

#### 3.2.1. Initial Work-Up

In all of these series, patients were re-staged with a clinical examination, and imaging at time of recurrence. Clinical examination was of utmost importance for mapping the local relapse and was performed under general or rachi anesthesia before starting the BT implantation. A metastatic work-up, with chest, abdomen, and pelvic CT scan, or positron emission tomography (TEP)-CT scan, was performed in every patient to rule out nodal or distant metastatic disease. Most of the centers performed pelvic MRI, in order to define the local extent of disease. Finally, the local relapse was histologically confirmed by biopsy.

#### 3.2.2. Type of Recurrence

The vast majority of the studies focused on isolated local recurrences, occurring in the vagina, paravaginal, pelvic sidewall, cervix (if hysterectomy was not performed as a part of the initial treatment), vulva or sub-urethra [31,32,33,34,35,36,37,38,39,40,41,42,43,45]. In only one study, regional and distant metastasis were not excluded, and occurred simultaneously to local relapse in four patients (two patients (9.1%) with nodal involvement and two patients (9.1%) with distant oligometastasis) [44].

Regarding the size of pelvic recurrences, the median tumor maximum diameter was reported in six studies [33,34,37,39,42,44], ranging from 20.5 to 36 mm (median, 24 mm), with the largest tumor diameter measuring up to 100 mm. Other studies reported median target (GTV or CTV) volumes, ranging from 6 to 50.35 cc (median, 31.4 cc) [32,41,45]. One study reported an average CTV volume of 60.9 cc, with the largest CTV volume up to 165.3 cc [31].

#### 3.2.3. Type of Primary Disease

Regarding primary disease, four studies included cervical cancer exclusively [33,36,40,41] and three studies included endometrial cancer [35,39,44]. In the remaining eight studies, several gynecologic malignancies were mixed up, with a majority of cervical and endometrial primaries and a few vaginal and/or vulvar primitives. In two studies, local recurrence of rectal, anal, and ovarian cancer were included, because of tumors re-occurring in previously irradiated fields [38,45] (Table 1).

#### 3.2.4. Previous RT Treatment

Among the 15 studies, 3 did not specify the previous radiation modalities [33,37,42]. In the remaining 12 studies, all patients received at least pelvic EBRT and/or BT. Prior RT treatment modalities are summarized in Appendix A.

Among the 12 studies with prior radiation data available, and 235 total patients recorded, the majority of patients (49.8%) had received EBRT + BT, 38.7% had been treated with EBRT exclusively, and 11.5% had received BT only. Previously received doses, when available, are detailed in Appendix A.

#### 3.2.5. Treatment-Free Interval

The interval time between first radiation and the recurrence was mentioned in nine studies and is reported as treatment-free interval (TFI) in Table 1 [32,33,35,39,40,41,43,44,45]. The median TFI ranged from 13 to 61 months (median, 23 months). Among these studies, three proposed re-irradiation if TFI was less than 6 months [41,43,45], three considered a minimal TFI of at least 6 months [32,39,40], and one of at least 1 year [44].

### 3.3. BT Specification

#### 3.3.1. Implantation Procedure

BT implantation was performed under general or epidural anesthesia. Transrectal ultra-sound guidance was often used, and sometimes per-procedure fluoroscopic guidance [31,32,33,34,37,38]. Gold seeds were sometimes used, placed in the superior, inferior, and lateral boundaries of the tumor [31,34]. Liu et al. performed interstitial implantation under real time 3D-CT guidance [36]. Interstitial BT was performed using a perineal template (Syed-Neblett template, Martinez Universal Perineal Interstitial Template (MUPIT), or institutional templates) or with free-hand needle application with a vaginal applicator. Position of the needles was anticipated using CT/MRI imaging and clinical examination. In case of endocavitary BT alone, a vaginal cylinder was used. Interstitial BT was performed in most of the studies, however intracavitary BT was reported in two studies, especially in the case of tumor thickness ≤5 mm [43,44].

#### 3.3.2. BT Technique

Feddock et al. used LDR BT exclusively, using ^131^Cs as an isotope, with doses ranging from 22 to 75 Gy [45]. In Amsbaugh and Aridgides et al.’s studies, some patients were treated with LDR BT (47.6% and 18%, respectively), both using Iridium-192 (^192^Ir), with a median prescription dose of 41.5 Gy (range, 28.5–55) and 30 Gy (range, 20–40), respectively [34,38]. Although these studies report on IGBT use, the limitations of LDR BT in terms of optimization abilities should be highlighted. In all remaining studies, HDR brachytherapy was used. In most cases, a twice daily (BID) HDR fractionation was used, with a median dose prescription of 29.4 Gy (range, 20–50), in 3 to 20 fractions, with a median dose per fraction of 5.3 Gy (range, 7–2, 5). In two studies, a once a day or weekly fractionation was reported: 10 to 15 daily fractions of 3 Gy or weekly 4 to 6 fractions of 5 to 7 Gy in Zolciak et al.’s study and 6 weekly fractions in Liu et al.’s work [36,43]. No study reported reirradiation using a PDR technique. BT techniques are detailed in Table 2.

#### 3.3.3. Treatment Planning Modalities

In all 15 studies, IGBT was performed with 3D treatment planning, however two studies included patients treated before the 2010′s through 2D treatment planning, accounting for 40% of the population (*n* = 8) in Zolciak et al.’s cohort [43], and for 77% in Mabuchi et al.’s study (*n* = 40) [33]. The majority of the studies used CT-guided BT, however MRI use was reported in three studies: six patients (20%) in the study from Mahantshetty et al. [40], 18 patients (27.3%) in Kamran et al.’s study [39], and 16 patients (72.7%) in Ling et al.’s cohort [44] had MRI-guided brachytherapy. Treatment planning modalities are detailed in Table 2.

#### 3.3.4. Target Delineation

There was some variability in the target contour definition and delineation among the analyzed studies. The GTV was delineated using CT and/or MRI and physical examination.

Most authors defined a high-risk clinical target volume (HR-CTV), with heterogeneous definitions [32,34,36,39,40,43,44]. Some have extrapolated the Groupe Européen de Curiethérapie and the European Society for Radiotherapy and Oncology (GEC ESTRO) recommendations published for cervical cancer target delineation [24,46], especially in the absence of prior hysterectomy [34,36,40,42,43]. For others, HR-CTV was defined as the GTV, plus a variable margin to account for the areas of high risk for microscopic spread to adjacent vaginal tissue, and was based on clinical suspicion of extension [32,44]. Liu et al. defined the HR-CTV from the GTV with an additional margin corresponding of the T2-weighted MRI gray zones [36]. Zolciak et al. defined the HR-CTV as the cervix and involved parametrium or the vaginal tumor and the paravaginal tissue, if involved, with a 1 cm margin in the longitudinal direction [43].

Some authors defined only a CTV, with CTV = GTV + variable additional margin (1 cm, or plus the upper third of vagina), excluding the OARs if uninvolved [31,33,35,37,38,41]. In Yoshida et al.’s study, in case of prior hysterectomy, CTV was equal to GTV, defined on T2 MRI [42].

An intermediate risk CTV was never used in the analyzed studies.

#### 3.3.5. Dose Constraints

Only few studies applied strict dose constraints on OARs and/or to the target volume, and most applied the ALARA (as low as reasonably achievable) principle, with OAR dose constraints determined case by case, while maintaining target coverage.

Nevertheless, in four studies [31,36,38,44], some dose constraints were suggested and are detailed in Table 3.

### 3.4. EBRT Doses and Techniques if Allowed before BT

In seven studies, EBRT preceded BT, with various dose schedules and in various proportions of the population, ranging from 15% to 66.7% [34,38,39,41,43,44,45].

Doses and techniques are synthetized in Appendix A. Interestingly, in three studies, reirradiation EBRT preceding BT was not allowed in case of prior EBRT + BT [34,38,44].

In case of prior EBRT + BT, reirradiation EBRT was performed with doses ranging from 20 to 30.6 Gy in 1.8 to 2 Gy/fraction in the central pelvis, sometimes with central shielding, not to compromise IGBT dose escalation, especially in case of multifocal relapse of the vagina, or large tumor volume [39,41,43,45]. In one study, inguinal nodes were irradiated in case of tumoral involvement of the inferior third of the vagina, at a dose of 45 to 50.4 Gy in 1.8 to 2 Gy/Fr because the area was never previously irradiated [44].

### 3.5. Dosimetry

Dosimetric data are detailed in Table 3, and when possible, doses are reported in 2 Gy equivalent doses per fractions (EQD2) applying the linear quadratic model with an α/β ratio of 10 Gy for tumor and 3 Gy for OARs, and a half-time repair of 1.5 h.

#### 3.5.1. Target

Doses delivered to target volume (HR-CTV, CTV, or GTV) are detailed in Table 3.

When EDQ2 doses were reported, median doses delivered to 90% of the HR-CTV (HR-CTV D_90_) or to 90% of the CTV (CTV D_90_) ranged from 39.4 to 72.2 Gy (median, 48.7 Gy; mean, 51.7 Gy) [31,33,36,38,39,40,41,42,43,44,45].

Some studies reported lifetime doses, taking into account prior RT. In Feddock et al.’s work, lifetime median EQD2 prescribed doses were 112.6 Gy (range, 75.7–144) at first salvage, and 152.2 Gy (range, 115.2–172.2) at second salvage [45]. Similarly, Kamran et al. reported a median cumulative EQD2 HR-CTV D_90_ of 78.1 Gy (range, 37–108.7) [39].

#### 3.5.2. OARs

Volumetric OAR dose constraints, such as the minimal dose delivered to the most exposed 2cc (D2cc) of normal tissue, were reported in most studies.

Data were easily accessible in studies where all the population were treated in a re-irradiation setting, however doses were not always converted in EQD2, making their interpretation and generalization difficult. Some studies reported OAR doses at time of reirradiation, without considering any prior RT, whereas others reported lifetime doses, taking into account previous RT doses. OARs doses are synthesized in Table 3.

In several studies mixing reirradiated and primo-irradiated patients, lifetime EQD2 OAR doses where sometimes available considering the reirradiated patients. As a result, Murakami et al. reported a median lifetime EQD2 rectum, bladder, and vaginal wall D2cc of 91.9 Gy (range, 71–114.3), 100.9 Gy (range, 69.7–120.3), and 170.1 Gy (range, 56.6–247.5), respectively [37].

### 3.6. Outcomes

Depending on the length of the follow-up, the rate of local control (LC), progression-free survival (PFS), disease-free survival (DFS), and overall survival (OS) were reported differently, at 3 months, at 2 years, at 3 years, and at 5 years, depending on the study (Table 1).

Rates of LC, PFS and/or DFS, and OS at 2 years ranged from 44% to 71.4% (median, 50.5%), from 20% to 42.9% (median, 40%), and from 52% to 78% (median, 60.15%) respectively, and were reported in four studies [31,32,40,41]. Rates of LC, PFS and/or DFS, and OS at 3 years ranged from 45% to 71% (median, 55.9%), from 40.8% to 52% (median, 42%), and from 42% to 68.1% (median, 61%) respectively, and were reported in four studies, including two studies combining patients with and without a history of prior RT, however results were reported separately in the reirradiated sub population [39,42,43,44]. In Martinez-Monge et al.’s study, rates of LC, PFS, and OS at 5 years were 71.4%, 21.4%, and 39.5% respectively, showing an excellent local control, but with high rates of regional and distant recurrences at 5 years [31].

### 3.7. Toxicity

The most reported type of toxicities were vaginal ulcers, necrosis, fistula (recto-vagina and/or bladder-vagina), urethral stenosis, rectal bleeding, cystitis, and pain (Table 1).

Grade 5 toxicity was reported in only one study, with one patient presenting post-implant bowel obstruction that caused her death one month after brachytherapy [31].

In this same phase II prospective trial from Martinez-Monge, rate of DFS without grade ≥3 at 2 toxicities at 2 years was the main objective and was at 40% [31].

In four studies, no grade 4 toxicity was reported [32,39,40,44]. Grade ≥3 toxicities ranged from 1.7% to 50% (median, 18.5%; mean, 21.85%) depending on the study [31,32,33,34,38,39,41,43,44,45].

### 3.8. Prognostic Factors

Among these 15 analyzed studies, some reported prognostic factors of outcome and/or toxicity. As a result, Mabuchi et al. showed on multivariate analysis that a tumor diameter ≥40 mm, a disease-free interval ≤6 months, and an International Federation of Obstetrics and Gynecology (FIGO) stage ≥III were predictors of shorter survival after re-RT BT (*p* = 0.0017, *p* = 0.015, and *p* = 0.042, respectively). In comparison to the palliative group, receiving best supportive care (BSC) only, the median post-recurrence survival was 31 months in the re-RT BT group, vs. 13 months (*p* < 0.0001). Nevertheless, survival rates were similar in both groups, among patients with two or more poor prognostic factors [33].

In Amsbaugh et al.’s study, a large tumor size was associated with a poorer PFS (a 1 cm increase in tumor size increased the hazard of an event (hazard ratio (HR), 1.62; 95% confidence interval (CI): 1.16–2.26)) [34]. In Umezawa et al.’s study, tumor size ≥4 cm was also associated with a worse local control (*p* = 0.009) [41]. Zolciak et al. also showed that a tumor diameter >3 cm was associated with poorer prognosis on OS, DFS, and LC (*p* = 0.001, *p* = 0.013, and *p* = 0.005, respectively), and that an interval between primary RT and reirradiation ≤12 months was associated with a poorer outcome (OS, DFS, and LC, with *p* = 0.001, *p* = 0.014, and *p* = 0.007, respectively) [43].

Mahantsetty et al. showed that local control rate was significantly higher in patients receiving >40 Gy EQD2 (52% vs. 34%, *p* = 0.05). Also, OS was better in patients without any poor prognostic factors, such as tumor size > 2 cm and a gap between the two courses of RT <25 months (2 years OS of 74%, vs. 50%, vs. 40%, for zero, one and two bad factors, respectively) [40]. Feddock et al. suggested that EQD2 reirradiation doses >45 Gy could be associated with a better outcome, with a 2 year failure rate of 57.1% vs. 29.3%, *p* = 0.07 [45].

Considering factors associated with greater toxicity, Umezawa et al. showed that a high volume receiving 100% (V100%) and 200% (V200%) of the prescribed dose was associated with a greater risk of grade ≥2 toxicities (52.2 vs. 21.6 cc for the V100%, *p* = 0.009, and 10.26 vs. 2.25 cc for the V200%, *p* = 0.001) [41]. Amsbaugh et al. suggested that urethra D_0_._1cc_ predicted for grade ≥2 urinary toxicity (1 Gy EQD2 increase = HR, 1.156; 95% CI: 1.001–1.335), however no dosimetric factors were found to predict vaginal or rectal toxicity [34].

Murakami et al. showed that an EDQ2 vaginal wall D2cc >145 Gy was predictive of vaginal toxicity, with a 2-year incidence rate of vaginal ulcer of 23.5% vs. 3.7%, *p* = 0.026 [37].

Finally, Kamran et al. showed that using MRI IGBT improved local control rate at 3 years, compared to CT IGBT (3-year LC rate of 100% vs. 78%, *p* = 0.04), with significantly fewer cumulative grade 3 urinary and rectal toxicities when using MRI IGBT [39].

## 4. Discussion and Highlights

Among these 15 studies, considerable heterogeneity was found in many aspects: BT dose schedules, modalities of prior RT (EBRT only, BT only, EBRT followed by BT), target volume and OAR dose reporting (physical doses, EQD2 doses, re-RT doses only, lifetime doses taking into account prior RT), and type of primary disease (endometrium, cervix, vagina, vulva, etc.), and some studies included a mixed population, with and without a past of prior radiation therapy, making the formulation of strict recommendations difficult in reirradiation IGBT in recurrent pelvic malignancies. Nevertheless, despite scarce data, based on small retrospectives and one prospective study with a small and heterogeneous population, some useful observations could be made, to assist and guide the clinician in this particular challenging situation.

First of all, these data showed that 3D-IGBT reirradiation in pelvic recurrences of gynecological malignancies is feasible, in highly selected patients not eligible for surgery due to medical contraindication, or in patients refusing surgery, and can be offered as an alternative treatment. Nevertheless, results in terms of outcome and toxicity are highly variable depending on studies, with a 2-year local control rate of only approximately 50%, in some studies [32,35,40,41] and with high frequency of grade ≥3 toxicities, up to approximately 50% in two studies [34,43], and one death reported within the month following the procedure, in one study [31].

On the contrary, some studies showed acceptable results, with a 3-year local control rate up to 70% [39,42,44] and with acceptable toxicities: less than 10% of grade ≥3 toxicities in four studies [32,38,42,44] and from 10% to 33% grade ≥3 toxicities in 8 others [31,33,35,37,39,40,41,45].

When comparing both modalities, the 3-year OS ranged from 42% to 68% in several re-RT IGBT studies and the 5-year OS was 39.5% in another, whereas 5-year survival rate varies between 10% and 56% in surgical studies [8,9,10,47,48,49,50,51]. Pelvic exenteration as salvage surgery, in previously irradiated tissues, comes at a high cost of complications, with associated morbidity ranging between 30% and 76%, high perioperative morbidity rates ranging between 22 and 66%, peri-operative mortality rates up to 5.5%, and with disabling long-term toxicities (fistula, bowel obstruction, need of permanent urinary and/or bowel derivations) [8,9,10,11,12,48,49,50,51,52]. As a result, a vast amount of patients experience long-term quality of life detriments, especially in the elderly [13].

Nevertheless, given the increased risks of complications inherent to the reirradiation setting, 3D-IGBT should be offered to a selected population of patients, and a clear and appropriate discussion of the risks and benefits of such procedure is essential. Every potential candidate of 3D-IGBT reirradiation should benefit from surgical expertise prior to treatment, to confirm the inoperability and to assess the eventual feasibility of colostomy or/and urinary diversion in case of future fistula/obstructions. The tolerance of previous irradiation should also be taken into account, and patients with major sequelae from their first irradiation course should probably be excluded.

Importantly, the time elapsed from prior RT should be considered, and IGBT reirradiation should not be offered if the recurrence occurs in an irradiated area within 6 months, after the first RT, and should be discussed with a disease-free interval between 6 to 12 months [33,43].

Also, it has been demonstrated that a recurrent tumor size from 2 to 4 cm constitutes a poor prognostic factor [33,40,41,43]; as a result, IGBT reirradiation should be clearly discussed in case of a recurrence size >4 cm. Finally, patients’ performance status, histology and grade of primary disease, and comorbidities exposing to higher risk of radio-induced toxicities (diabetes, smoking, severe hypertension, inflammatory bowel disease) [53] should also be considered before proceeding [33,34,37,43,54,55].

Determining the ideal dose to target volume and dose constraints for OARs is complex. If Manhanstetty et al. found that an EQD2 dose >40 Gy was associated with a better LC [40], Ling et al. failed to prove that an EQD2 HR-CTVD_90_ was associated with a better outcome, by lack of significance (3-year LC of 78.8% if HR-CTVD_90_ > 65 Gy, vs. 50.5% otherwise, *p* = 0.48) [44]. Interestingly, assessing ultrasound-guided 2D-planned IS-BT in vaginal recurrences from cervical and endometrial carcinomas, Weitman et al. showed that the association of an EQD2 prescribed total dose >64 Gy with a good coverage index (>0.8) was a significant prognostic factor for remission status 3 months after RT (*p* = 0.0419), for a better LC (*p* = 0.0286) and a better OS (*p* = 0.0121) [54].

In all 15 studies, no association could be made between doses and OARs (bladder, rectum, sigmoid) and the risk of late toxicity, probably due to the small number of patients treated in these series. Nevertheless, in the study with the higher rate of grade ≥3 late toxicities, a cumulative EDQ2 of approximately 100 Gy was delivered to the rectum and bladder D2cc [43].

In light of these considerations, we recommend that doses to the target should be at least 40 Gy EQD2, and two different approaches could be considered, as mentioned in the American Brachytherapy Society (ABS) working group report of reirradiation for gynecological cancers [28]. The first approach, taking prior RT into account, is to exceed the cumulative classical tolerance doses to OARs (e.g., those recommended in the image guided intensity modulated External beam radiochemotherapy and MRI based adaptive BRAchytherapy in locally advanced CErvical cancer (EMBRACE) II study: bladder EQD2 D2cc of 80–90 Gy, rectum and sigmoid EQD2 D2cc of 70–75 Gy) [56] in order to approach a 60–65 Gy to the HR-CTVD_90_. The second is to strictly respect these OARs constraints, aiming to maintain a HR-CTVD_90_ >40 Gy. In the first strategy, patients must be appropriately informed about the potential risk of late side effects, with a clear and loyal consent. Indeed, the risk of severe complications seems significant with a lifetime EDQ2 bladder and rectum D2cc of 100 Gy. Such threshold may be difficult, however, to respect, if the recurrence occurs in an area previously exposed to full-dose radiotherapy (45–50 Gy). One difficulty is to have a precise estimation of the total dose delivered to D2cc, as treatment planning systems cannot sum the doses delivered by each irradiation, in the context of geometrical changes (an arithmetic sum of the D2cc is overestimating the true organ exposure, as the D2cc may not be at the same anatomic level).

Considering the place of EBRT preceding 3D-IGBT in the reirradiation setting, such strategy was proposed in several studies, especially in cases of multifocal vaginal recurrence, nodal involvement at recurrence, and of course, in cases of exclusive BT in the first radiation course [34,38,39,41,43,44,45]. Such strategy could be proposed, with a normo-fractionated 45 Gy pelvic EBRT in the absence of prior EBRT. If prior treatment consisted in EBRT ± BT, in case of multifocal vaginal recurrence, or in case of tumor size > 2–4 cm, EBRT could be discussed for shrinking the tumor, making it more accessible to IGBT. Nevertheless, highly conformal techniques and small radiation fields should be used, in a normo-fractionated (or hyper-fractionated) dose range of 30–36 Gy and EBRT should not compromise IGBT dose escalation. Indeed, the respective contributions of EBRT and BT must be carefully weighted, and it always seems preferable to prioritize BT (if feasible) in order to spare OARs.

Regarding the target delineation, most of the studies used a HR-CTV, based on imaging and physical examination plus a variable margin, taking into account the areas of high risk of microscopic spread to the adjacent tissue. In cases of cervical or parametrial recurrences, HR-CTV could be extrapolated from the GEC ESTRO recommendations for definitive cervical cancer target delineation [24,46]. In cases of vaginal recurrences, HR-CTV could be built based on the GEC ESTRO target contouring recommendations for primary vaginal cancer [57]. In this reirradiation setting, IR-CTV delineation could be discussed, but was not used in the analyzed series.

In the vast majority of the analyzed studies, HDR IGBT was used, and both clinical outcomes and grade ≥3 toxicities seemed better in recurrent endometrial cancer series than in recurrent cervical cancer series (Table 1). One explanation could be that prior RT in cervical cancer consisted mainly in EBRT + BT, thus limiting the dose delivered to the target, while greater doses were received by the OARs. In Feddock et al.’s study, reirradiation using LDR offered a good local control (73% at last follow-up or death) with few grade ≥3 toxicities (16.7%) and no grade 5 [45]. The radiobiological advantages of permanent interstitial brachytherapy have been described in limiting normal tissue toxicity compared to HDR BT, however dose delivery is operator-dependent, with a need of an accurate source placement and spacing, without the optimization possibility [58,59]. In this way, PDR BT could potentially offer a better therapeutic index than HDR, allowing some repair in late-reacting normal tissue due to intervals between pulses, with the advantage of isodose optimization and radiation safety, compared to LDR [60,61]. Unfortunately, none of the analyzed studies used this technique, although it should be more widely used, especially in the reirradiation setting.

In this complex clinical setting of reirradiation, efforts should be made to expand the therapeutic index. MRI-IGBT, allowing an accurate delineation of the gynecologic tumor and normal pelvic structures, has shown its benefit in cervical and vaginal cancer in terms of OARs sparing and dose escalation to the tumor [24,62,63,64]. As suggested by Kamran et al., MRI-IGBT could be preferred to CT-IGBT in the reirradiation setting, with better outcomes and less toxicities in a subgroup analysis of few patients [39].

Another way to reduce the doses delivered to normal tissue, while facilitating dose escalation within the tumor, could be the use of hydrogel spacers, as described by Viswanathan et al. in their short series of three patients with locally recurrent gynecologic malignancies requiring reirradiation [65]. This hydrogel, inserted between the rectum and the posterior wall of the vagina through ultrasound guidance, safely separated rectum and small bowel from the tumor, and resulted in significant lower radiation doses administered to rectum, small bowel, and sigmoid. Larger and prospective studies are warranted to confirm the clinical impact of such technique.

Finally, future investigation should combine reirradiation 3D-IGBT with tumor-specific radio sensitizing agents. Gadolinium-based nanoparticles AGuiX^®^ are able to specifically penetrate tumors, can act as contrast agents, and are used as a radio-sensitizing agent by locally increasing the energy deposited by the incident radiation [66]. This nanoparticle has been investigated recently in association with radiotherapy in a phase I trial in brain metastasis [67] and a phase I trial is ongoing in locally advanced cervical cancer in association with brachytherapy (NANOCOL, NCT03308604). Further investigation could assess this radio-sensitizing agent with reirradiation MRI-guided BT in local recurrences of gynecologic cancer to improve local control while limiting the risk of toxicities.

Figure 2 and Figure 3 illustrate two clinical examples of reirradiation with MRI-guided PDR BT performed in our center.

## 5. Conclusions

Reirradiation 3D-IGBT seems to be feasible and may be offered as an alternative to salvage surgery in selected cases of inoperable patients or for those refusing surgery, in the context of pelvic recurrences of gynecologic malignancies. Nevertheless, results in terms of outcome and toxicity are highly variable depending on studies, with, in some cases, a high risk of grade ≥3 late toxicity (up to 50% in 2/15 studies), and should encourage caution. We recommend that such patients be referred to centers that treat a high volume of gynecological malignancies, as the experience of the physician may be an important factor in the management of these patients. By analogy to 3D-IGBT in cervical or vaginal cancer, MRI-guided BT might be preferred in this context of reirradiation. Due to the heterogeneity and the small size of populations reported in the studies, no formal conclusions or strict recommendations could be made, especially regarding the doses required to offer the best local control and the dose constraints applicable to the OARs. Each case must be thoughtfully discussed, to offer the best personalized treatment, in this particularly difficult clinical setting. Centralization of data and large-scale multicentric international prospective trials are warranted. Efforts should be made to improve local control while limiting the risk of toxicities. Innovative techniques and/or treatment (e.g., hydrogel spacers or novel approach of radio-sensitizing agents) should be prospectively assessed, to expand the therapeutic index.

## Figures and Tables

**Figure 1 cancers-13-01226-f001:**
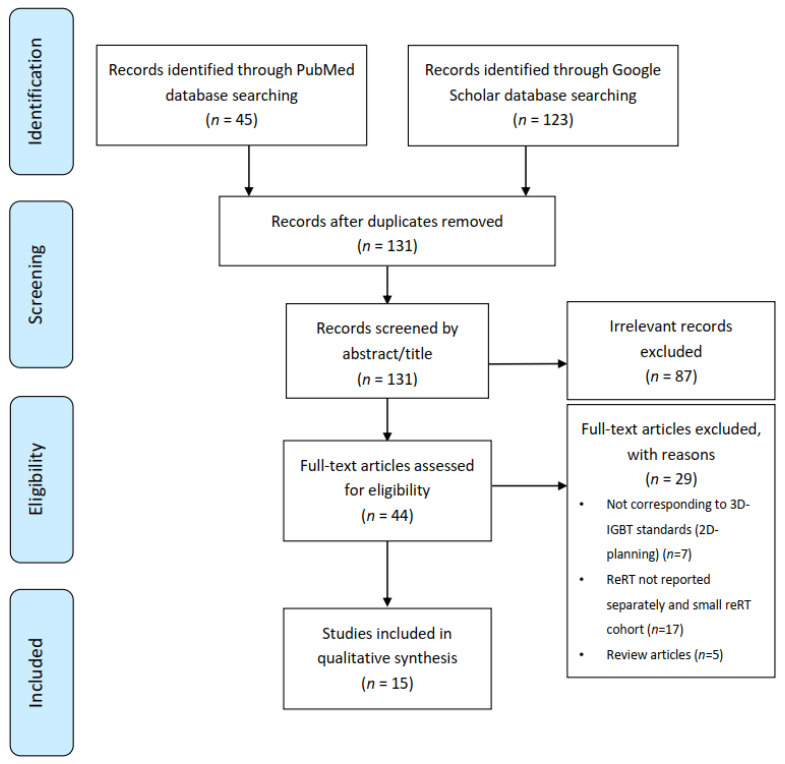
Data selection process: Preferred Reporting Items for Systematic Review and Meta-Analyses (PRISMA) flow diagram.

**Figure 2 cancers-13-01226-f002:**
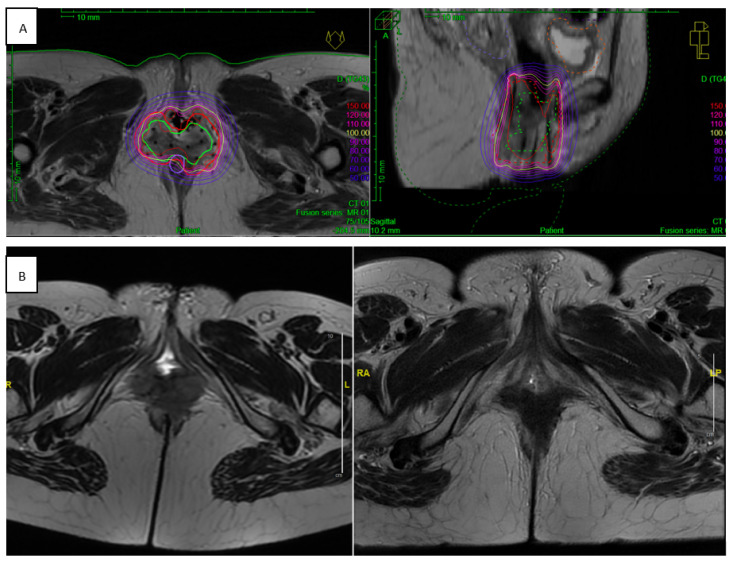
Example of MRI-guided PDR BT reirradiation for vulvo vaginal recurrence. (**A**) MRI-guided PDR BT reirradiation implant to a 6 cm vulvo-vaginal recurrence of cervical epidermoid carcinoma, initially treated in another center with pelvic EBRT (45 Gy in 25 fractions) followed by HDR BT (25 Gy in 5 fractions of 5 Gy). Because of the size (>3 cm) of the vaginal recurrence, the patient was treated first by pelvic EBRT (39.6 Gy in 22 fractions of 1.8 Gy in the tumor, and 45 Gy in 25 fractions of 1.8 Gy in inguinal lymph nodes), followed by an MRI-guided PDR BT with vaginal mold associated with free-hand needles, at the dose of 20.10 Gy in 67 pulses of 0.30 Gy. The EQD2 reirradiation doses (EBRT + BT) were the following: HR-CTV D_90_ = 67.42 Gy, bladder D2cc = 42.73 Gy, rectum D2cc = 60.57 Gy, sigmoid D2cc = 39.47 Gy, and small bowel D2cc = 39.11 Gy. (**B**) Pelvic MRI showing the vulvo-vaginal recurrence before re-RT (left), and a complete response 8 weeks after MRI-guided PDT BT (right).

**Figure 3 cancers-13-01226-f003:**
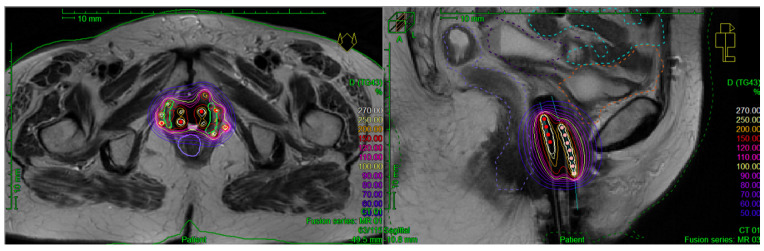
Example of MRI-guided PDR BT reirradiation for vaginal recurrence. MRI-guided PDR BT reirradiation implant to a bi-focal recurrence of endometrial adenocarcinoma to the lower third of the vagina, initially treated by total hysterectomy, followed by pelvic EBRT (45 Gy in 25 fractions) and by HDR BT of vaginal cuff (10 Gy in 2 weekly fractions of 5 Gy). Because of the bi-focality of the vaginal recurrence, the patient was treated first by vaginal EBRT (30.6 Gy in 17 fractions of 1.8 Gy), followed by MRI-guided PDR BT with vaginal mold associated with free-hand needles, at the dose of 30 Gy in 100 pulses of 0.30 Gy. The EQD2 reirradiation doses (EBRT + BT) were the following: HR-CTV D_90_ = 73.19 Gy, bladder D2cc = 39.76 Gy, rectum D2cc = 43.85 Gy, sigmoid D2cc = 32.98 Gy, and small bowel D2cc = 31.97 Gy.

**Table 1 cancers-13-01226-t001:** Reirradiation three-dimensional image-guided brachytherapy (3D-IGBT) studies: general aspects, outcome, and toxicity.

Study	Type of Study	No. of Reirradiated Pts/No. of Pts Included	Primary Disease	Treatment Free Interval (Range)	Median Follow-Up	Outcome	Toxicity
Raziee, 2020 [32]	R	26/26 (100%)	Endometrium (76.9%) Cervix (15.3%) Vagina (3.9%) Vulva (3.9%)	20.3 months (9.9–30.5)	24 months (12.7–30.8)	2-year LC 50% 2-year PFS 38% 2-year OS 78% CR: 65%	No gr 4–5 Gr 3: 7.7%, Gr 1–2: 38.5%
Ling, 2019 [44]	R	22/22 (100%)	Endometrium	26.6 months (17.2–54.4)	27.6 months	3-year LC 65.8% 3-year DFS 40.8% 3-year OS 68.1%	No gr 4–5 Gr 3: 4.5% (ureteral stricture)
Umezawa, 2018 [41]	R	18/18 (100%)	Cervix	14.9 months (3.1–53.6)	18.1 months (8.4–75.4)	2-year LC 51% 2-year PFS 20% 2-year OS 61% CR: 66.6% PR: 33.3%	No Gr 5 Gr ≥ 3: 17% Gr 4 vagina fistula: 11.1% Gr 3 vagina fistula: 5.5%
Martinez- Monge, 2014 [31]	Phase II	15/15 (100%)	Endometrium (40%) Cervix (40%) Vagina (20%)	NR	2.9 years (1.2–9.2)	2-year DFS without gr ≥ 3 toxicity 40% 2-year LC 71.4%, 5y LC 71.4% 2-year PFS 42.9%, 5y PFS 21.4% 2-year OS 59.3%, 5y 0S 39.5%	Gr 5: 6.6% (*n* = 1, bowel obstruction) Gr ≥ 3: 20%
Feddock, 2017 [45]	R	42/42 (100%)	Cervix (28.6%) Endometrium (26.2%) Vagina (21.4%) Vulva (11.9%) Fallopian tube, rectal, anal (11.9%)	26.1 months (1.4–8.1 years)	16.3 months	73% LC at death 52% OS at last follow-up	No Grade 5 Gr ≥ 3: 16.7% (vaginal necrosis: 9.5%, vaginal fistula: 7.1%)
Liu, 2016 [36]	R	16/16 (100%)	Cervix	NR	NR	At 3 months: CR: 37.5%, PR: 43.8%, SD: 18.7%, No PD	NR
Mahantshetty, 2014 [40]	R	30/30 (100%)	Cervix	25 months (7–227)	25 months (3–96)	2-year LC 44% 2-year DFS 42% 2-year OS 52% CR: 76%, PR: 18%, SD: 3%, PD: 3%	No gr 4–5 Gr 3 rectal & urinary: 10% Gr 3 vagina: 10% Gr 2 small bowel: 10%
Zolciak-Siwinska, 2014 [43]	R	20/20 (100%)	Cervix (70%) Vagina (30%)	23 months (3–76)	31 months	3-year LC 45% 3-year DFS 42% 3-year OS 68% CR: 95%, LR: 45%, DR: 45%	No gr 5 Gr 3: urinary10%, Gi 5% Gr 3–4 vagina: 40% (obliteration of vagina)
Mabuchi, 2014 [33]	R	52/52 (100%)	Cervix	13 months	55.6 months	5-year OS 52.6% CR: 59.6%, PR: 17.3%, SD: 15.4% PD: 53.8%, LR: 35.7%, RR: 17.9% (pelvic sidewall) DR: 39.3%, DR + LR: 7.1%	No gr 5 Gr ≥ 3: 25% (fistula: 17.3%)
Kamran, 2017 [39]	R	24/66 (36%)	Endometrium	20 months	33 months	3-year LC 71% 3-year DFS 52% 3-year OS 54%	Gr 3: 33% (urinary 16.7%, rectal 25%) No gr 4–5
Murakami, 2016 [37]	R	8/44 (18%)	Cervix	NR	NR	3-year LC 45% 3-year OS 51%	Gr 4: 27% (vagina fistula) Gr 2: 12.5% (vagina necrosis)
Amsbaugh, 2015 [34]	R	18/21 (86%)	Endometrium (52.4%) Cervix (33.3%) Vulva (14.3%)	NR	16.5 months	1-year LC 71.5% 1-year PFS 66% 1-year OS 82.2% 2-year OS 52.5% 52.4% relapses LR: 23.8%, RR: 14.3%, DR: 14.3%	No gr 4–5 Gr 3: Vaginal (28.6%) urinary (9.5%), rectal (19%) Gr 1–2: Vaginal (71.4%), urinary (47.6%), rectal (23.8%)
Yoshida, 2015 [42]	R	21/56 (38%)	Cervix (80.4%) Endometrium (19.6%)	NR	33 months	Initial radical hysterctomy + adjuvant RT: 3-year LC 75% 3-year OS 88% Initial definitive RT: 3-year LC 46% 3-year OS 42%	Gr ≥3: 4.8% (vaginal fistula)
Aridgides, 2016 [38]	R	6/60 (10%)	Endometrium (26.7%) Cervix (8.3%) Vulva (5%) Ovary (3.3%) Rectal (1.7%)	NR	36.9 months (4–234)	Overall LC: 66.7% Local Failure: 33.3% Distant Relapse: 33.3%	Gr 3 soft tissue necrosis: 1.7%
Huang, 2016 [35]	R	16/40 (40%)	Endometrium	61 months	18 months	2-year LC 53% 2-year PFS 44% 2-year OS 67%	No Gr 5. Gr 1–2: 52.2% Gr ≥ 3: 12.5% (Rectal bleeding, Rectovaginal fistula, Radiation necrosis, cystitis)

Legend: R = retrospective; CTV = clinical target volume; GTV = gross tumor volume; HR-CTV = high-risk clinical target volume; OAR = organ-at-risk; HDR = high dose rate brachytherapy; LDR = low dose rate brachytherapy; EQD2 = 2 Gy equivalent doses per fractions; NR = not reported; LC = local control; PFS = progression-free survival; DFS = disease-free survival; LR = local relapse; RR = regional relapse (i.e., lymph nodes); DR = distant relapse; CR = complete response; PR = partial response; SD = stable disease; PD = progression disease; GI = gastro-intestinal.

**Table 2 cancers-13-01226-t002:** Reirradiation 3D-IGBT studies: BT and EBRT technique and doses.

Study	BT Technique: HDR/LDR	BT Technique: IC/IS	BT Technique: Dose Schedule (Physical Dose)	Treatment Planning: 2D-Planning Allowed?	3D-IGBT: MRI/CT
Raziee, 2020 [32]	HDR	IS	Median 30 Gy, in 3 to 6 Fr, BID Median HR-CTV D90: 29.1 Gy (16.1–64.6)	No	CT-planned
Ling, 2019 [44]	HDR	IC and IS (IC if tumor thickness ≤ 5 mm)	Median 28.75 Gy (range, 24.8–30) In 4 to 7 Fr, BID	No	CT- and MRI-planned (MRI in 16 patients (72.7%))
Umezawa, 2018 [41]	HDR	IS	Median 48 Gy (range, 24–50) in 8 Fr (range, 4–20), BID	No	CT-planned
Martinez-Monge, 2014 [31]	HDR	IS	38 Gy in 8 Fr, BID	No	CT-planned
Feddock, 2017 [45]	LDR	IS	Template-guided median dose: 47.5 Gy (range, 25–55) Free-hand median dose: 50 Gy (range, 22–75)	No	CT-planned
Liu, 2016 [36]	HDR	IS	36 Gy in 6 Fr	No	CT-planned
Mahantshetty, 2014 [40]	HDR	IS	3 to 4 Gy/Fr, in 6 to 13 Fr, BID	No	CT- and MRI-planned (MRI in 6 patients (20%))
Zolciak-Siwinska, 2014 [43]	HDR	IC and IS (IC if tumor thickness ≤5 mm)	10 to 15 daily Fr of 3 Gy, or 4 to 6 once a week Fr of 5 to 7.5 Gy	Yes 8 patients (40%)	CT-planned
Mabuchi, 2014 [33]	HDR	IS	42 Gy in 7 Fr, BID	Yes 40 patients (77%)	CT-planned
Kamran, 2017 [39]	HDR and LDR	IS	Median 22.5 Gy in 5 Fr, BID	No	CT- and MRI-planned (MRI: 18 patients (75%))
Murakami, 2016 [37]	HDR	IS	Median 32 Gy (range, 36–48), Median dose/Fr 5.3 Gy (range, 4–6) *	No	CT-planned
Amsbaugh, 2015 [34]	HDR and LDR	IS	HDR: Median 22.5 Gy (range, 13.5–30), 3 to 5 Fr, BID LDR: Median 41.5 Gy (range, 28.5–55)	No	CT-planned
Yoshida, 2015 [42]	HDR	IS	Median 48 Gy (range, 42–51), in 7 to 8 Fr, BID	No	CT-planned
Aridgides, 2016 [38]	HDR and LDR	IS	HDR: Median 20 Gy (range, 13.5–30), in 4 Fr, BID LDR: Median 30 Gy (20–40)	No	CT-planned
Huang, 2016 [35]	HDR	IS	Median 21 Gy (range, 15–27.5), in 2 to 5 Fr	No	CT-planned

Legend: BT = brachytherapy; MRI = magnetic resonance imaging; CT = computed tomography; HDR = high dose rate brachytherapy; LDR = low dose rate brachytherapy; IC = intra-cavitary; IS = interstitial; BID = twice daily; EBRT = external beam radiation therapy; IGBT = image-guided brachytherapy; Fr = fraction; NR = not reported. *: If not reported separately for patients receiving reirradiation.

**Table 3 cancers-13-01226-t003:** Reirradiation 3D-IGBT dosimetric details and tumor volume.

Study	Doses to Target Volume (EQD2)	Doses to OARs	Tumor Volume	Dose Constraints
Raziee, 2020 [32]	NR	Not considering prior RT Bladder D2cc: 15.5 Gy (11–23.2) Rectum D2cc: 18.7 Gy (15.8–22.5) Sigmoid D2cc 3.7 Gy (1.9–5.6)	HR-CTV in cm^3^: 34.6 (4.8–96)	No
Ling, 2019 [44]	Median prescribed dose 56 Gy Not considering prior RT CTV D_90_ 72.2 Gy (63.9–78.1)	Not considering prior RT Bladder D2cc: 54.3 Gy (EQD2) (23.9–93.4) Rectum D2cc: 50.6 Gy (EQD2) (34.6–82)	Median tumor diameter 23 mm (3–33)	Lifetime bladder D2cc (EQD2) < 90 Gy Lifetime rectosigmoid D2cc (EQD2) < 75 Gy Ideal HR-CTV D_90_ (EQD2) > 60 Gy
Umezawa, 2018 [41]	Not considering prior RT HR-CTV D_90_: 62.6 Gy (48.6–82.5)	Not considering prior RT Sigmoid D2cc: 39.9 Gy (EQD2) (3–60) Rectum and bladder D2cc: 36.8 Gy (EQD2) (12–62)	Median CTV volume 50.35 cm^3^ (2.1–129.2)	No
Martinez-Monge, 2014 [31]	Median prescribed dose to CTV D_90_: 46.7 Gy Not considering prior RT CTV D_90_: 43 Gy ± 7.4 GTV D_90_: 54.2 Gy ± 9.4	Not considering prior RT Bladder D2cc: 37.4 Gy (EQD2) ± 15.1 Rectum D2cc: 27.5 Gy (EQD2) ± 8.5 Lifetime EQD2: Bladder D2cc: 120.8 Gy ± 50 Rectum D2cc: 110.8 Gy ± 48.2	Average CTV volume 60.9 cm^3^ (14.8–165.3)	Average urethral dose < 5.5 Gy/Fr (<15% of prescription dose) Rectal D_10_ < 3.3 Gy (<70% of the prescription dose) Bladder D_10_ < 3.8 Gy (<80% of the prescription dose)
Feddock, 2017 [45]	Not considering prior RT -At first salvage: median 44.3 Gy (17.1–81.5) -At 2^nd^ salvage: median 39.4 Gy (21.5–49.1) Lifetimes doses: -At first salvage: median 112.6 Gy (75.7–144) -At 2nd salvage: median 152.2 Gy (115.2–172.2)	Not considering prior RT Rectum D2cc: 27.8 Gy (12.9–43.6) Bladder D2cc: 34.2 Gy (21.3–46.1).	-Free-hand implant: Median GTV: 6 cm^3^ (2.25–44.5) -Template-guided implant Median GTV: 27.9 cm^3^ (12.2–44.6)	No
Liu, 2016 [36]	Median prescribed dose to the HR-CTV: 48 Gy Not considering prior RT Median HR-CTV D_90_: 52.5 ± 3.3 Gy	Taking prior RT into account Median bladder D2cc: 85.6 Gy (EQD2) ± 5.8 Gy Median rectum D2cc: 71.6 Gy (EQD2) ± 6.4 Gy Median sigmoid D2cc: 69.6 Gy (EQD2) ± 5.9 Gy	NR	HR-CTV D_90_ (EQD2) ≥ 50 Gy Bladder D2cc (EQD2) ≤ 90 Gy Rectum D2cc (EQD2) ≤ 75 Gy Sigmoid D2cc (EQD2) ≤ 75 Gy
Mahant-shetty, 2014 [40]	Median prescription dose: 42 Gy (31.5–54.4)	Not considering prior RT (DVH available n = 6) Mean bladder D2cc: 42.1 Gy (EQD2) ± 13.3 Mean rectum D2cc: 25.1 Gy (EQD2) ± 8.9 Mean sigmoid D2cc: 25.1 Gy (EQD2) ± 12.1	Median tumor diameter: <20 mm: 46% (*n* = 14) 20–40 mm: 40% (*n* = 12) >40 mm: 14% (*n* = 4)	No
Zolciak-Siwinska, 2014 [43]	Median prescription dose: 48.8 Gy (16–91) Not considering prior RT Median D_100_ 31.75 Gy (10–69.8) Median lifetime prescribed dose 133.5 Gy (96.8–164.2)	Not considering prior RT Median rectum D2cc: 42.4 Gy (EQD2) (7.4–78.8 Gy) Median bladder D2cc: 42.7 Gy (EQD2) (6.4–84.8 Gy) Median lifetime doses at OARs, if 3D planned: Rectum D2cc: 94.4 Gy (EQD2) (67.1–118.8 Gy) Bladder D2cc: 99.3 Gy (EQD2) (70.4–122.3) Median lifetime doses at OARs, if 2D planned: Rectal point: 119.75 Gy (EQD2) (82–138.3) Bladder point: 113.5 Gy (EQD2) (75–139)	Median tumor diameter: ≤30 mm: 60% (*n* = 12) >30 mm: 40% (*n* = 8)	No
Mabuchi, 2014 [33]	Median prescribed dose: 56 Gy Not considering prior RT CTV D_90_: 72.2 Gy (63.9–78.1)	Not considering prior RT Bladder D2cc: 54 Gy (EQD2) (23.9–93.4) Rectum D2cc 50.6 Gy (EQD2) (34.6–82)	Median tumor diameter 22.5 mm	No
Kamran, 2017 [39]	Not considering prior RT: Median HR-CTV D_90_ 41.8 Gy (10.4–77.3) Lifetime doses: Median cumulative dose to HR-CTV D_90_ 78.1 Gy (37–108.7 Gy)	NR	Median tumor diameter 20.5 mm (5–84)	No
Murakami, 2016 [37]	Mean prescribed dose 54 Gy (42–64)	Lifetime doses: Rectum D2cc: 91.1 Gy (EQD2) (71–114.3) Bladder D2cc: 100.9 Gy (EQD2) (69.7–120.3) Vaginal wall D2cc: 170.1 Gy (EQD2) (56.6–247.5)	Median tumor diameter 36 mm (10–80)	No
Amsbaugh, 2015 [34]	NR	NR	Median tumor diameter 30 mm (15–100)	No
Yoshida, 2015 [42]	Median planning aim dose 64 Gy	NR	Median tumor diameter 25 mm (5–79)	No
Aridgides, 2016 [38]	Median prescribed dose 48.7 Gy	NR	NR	Rectum Dmax (EQD2) < 70 Gy Urethra Dmax (EQD2) < 70 Gy Bladder Dmax (EQD2) < 75 Gy
Huang, 2016 [35]	NR	NR	NR	No

Legend: CTV = clinical target volume; GTV = gross tumor volume; HR-CTV = high-risk clinical target volume; OAR = organ-at-risk; HDR = high dose rate brachytherapy; LDR = low dose rate brachytherapy; EQD2= 2 Gy equivalent doses per fractions; NR = not reported; GI = gastro-intestinal; D2cc = minimal dose delivered to the most exposed 2cc of normal tissue; HR-CTV D_90_ = dose delivered to 90% of the HR-CTV; CTV D_90_ = dose delivered to 90% of the CTV; GTV D_90_ = dose delivered to 90% of the GTV.

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
