# Peer review of "Image-Guided Brachytherapy for Salvage Reirradiation: A Systematic Review"

_cancers, 2021, doi:10.3390/cancers13061226_

Round 1

Reviewer 1 Report

Distinction between cervical cancers and other cancers seems important since the majority of cervical carcinomas are squamous type and HPV related.

Reviewer 2 Report

In this manuscript, the authors present a literature review on image-guided adaptive brachytherapy (IGABT) in the reirradiation of gynecological neoplasms. The authors made a great analytical effort to extract a large amount of data from the selected papers. The conclusions of their analysis are:

1) IGABT is an effective alternative to surgery

2) IGABT is feasible

3) MRI planned IGABT should be preferred.

However, the results in terms of outcome are highly variable, in some cases certainly not optimal (in some cases 2-year local control was only about 50%). In addition, grade> = 3 toxicity reached 50% of patients. Finally, only 40 patients included in the analysis received MRI-planned IGABT, and the supposed superiority of the latter is based on a subgroup of only 18 patients from the study by Kamran et al.

For these reasons, I believe that the manuscript's conclusions are not justified by the presented data.

To this can be added a number of problems:

  • The quality of the language and syntax is poor due to typos, grammatical errors, and the frequent use of "unusual" terms ("planification", "drastic regional and distant recurrences", "patients recused", "was to be deplore ", "a bad prognosis factor ")
  • Some expressions seem clearly contradictory "100 Gy could be safely delivered ... but at a cost of 50% of grade 3 toxicities"
  • The purpose of the paper is to describe the results of adaptive brachytherapy, but a clear definition of this term is missing from the manuscript. On the contrary, it would be essential to clearly define the concept of IGABT right from its introduction, also in consideration of the multiple meanings of "adaptive radiotherapy"
  • The authors define their analysis as "systematic". However, no specific guidelines were followed for this type of analysis (for example: PRISMA guidelines). Furthermore, the search was performed on a single bibliographic database (PubMed) and the terms used in the search are in some cases questionable (for example: "repeat radiation therapy" "second radiotherapy" "cervix cancer")
  • The tables are poorly legible. In particular, table 1 is 12 pages long, which prevents any "synoptic" view of methods and results. Perhaps the authors could have simplified the tables to what is strictly necessary and possibly added more detailed analysis in the supplementary materials. Moreover, there are evident inhomogeneities in the abbreviations (for example "Grade" or "Gr" in the same table)
  • The discussion, which after such a detailed analysis should help readers make a summary, is quite long and in part simply repeats what is contained in the results. Furthermore, in the discussion, the authors include a long paragraph to describe the technique used in their center, which is scarcely relevant with an analysis of the literature.

Ultimately, I believe that the manuscript in its present form cannot be published. I would advise the authors to review the entire text, define the IGABT concept from the beginning, simplify the tables, simplify the discussion, carefully review the entire text, possibly with the help of a native English speaker, and above all to harmonize the conclusions of the manuscript with the results of the analysis.

Reviewer 3 Report

The manuscript describes a systematic review of image-guided brachytherapy for salvage re-irradiation in gynecological cancer. 

The review is generally well written and the discussion and conclusions will be of interest.  However, there are a number of issues that should be addressed.

The Simple Summary needs to be re-written – at the moment it is just a shortened version of the Abstract. Please re-write using lay person’s terms.

In the Materials and Methods section, please give more information on what makes this a ‘systematic review’ rather than just a literature review, for example, please state more clearly the study selection criteria and the reasons for inclusion/exclusion. The authors mention they searched the PubMed database but for a systematic review, multiple resources should be searched. Please ensure that all relevant papers have been captured for the review and included if appropriate.

In the Conclusions, the authors state ‘Future investigations should combine IGABT with a novel approach of radio-sensitizing agents (NANOCOL, NCT03308604), to improve local control while limiting the risk of toxicities and could be assessed in the recurrence setting.’  This statement seems to appear at random as the authors have not mentioned radio-sensitizing agents at all up to this and it is not clear why this particular one, NANOCOL, NCT03308604, is mentioned. More text needs to be added on this earlier in the manuscript or this part of the Conclusion should be removed.

Table 1, 2 and 3, please re-format the tables to make them easier to read and to flow better with the text (ie. not to have to go back and forward through each table when reading through the text).

Although the review is generally well written, language and grammar etc needs to be improved in places.

Round 2

Reviewer 2 Report

I commend the authors for their great effort to improve their manuscript. In particular, the tables are very clear and the structure of the paper is formally that of a systematic review. Unfortunately, it is difficult to accurately evaluate all the modifications made by the authors, since many parts of the text seem to have been deleted and replaced by an almost identical text. In other words, it would help the reviewer a lot to be able to identify as "tracked" only the parts that were actually modified. Furthermore, I believe that the figure on the selection of papers can be replaced in the text with the "official" one of the PRISMA, now included in the supplementary material.

Author Response

I commend the authors for their great effort to improve their manuscript. In particular, the tables are very clear and the structure of the paper is formally that of a systematic review.

Unfortunately, it is difficult to accurately evaluate all the modifications made by the authors, since many parts of the text seem to have been deleted and replaced by an almost identical text. In other words, it would help the reviewer a lot to be able to identify as "tracked" only the parts that were actually modified.

Furthermore, I believe that the figure on the selection of papers can be replaced in the text with the "official" one of the PRISMA, now included in the supplementary material.

Response 1: We thank the reviewer for their relevant comment and apologize for the inconvenience. We identified as “tracked” the modified parts and hope it would suit you.

Figure 1 has been changed by the PRISMA flowchart, initially identified as supplementary material.

Reviewer 3 Report

The authors have satisfactorily addressed all my concerns.

However, English grammar needs to be corrected throughout the manuscript.

Author Response

The authors have satisfactorily addressed all my concerns.

However, English grammar needs to be corrected throughout the manuscript.

Response 1:

We thank the reviewer for their relevant comment. We performed another English editing by an English native speaker and hope this will suit you.

Round 3

Reviewer 2 Report

The authors of the manuscript have further improved the quality of the latter and simplified the possibility of evaluating the changes made.